# Epidemiology of congenital heart defects in France from 2013 to 2022 using the PMSI-MCO (French Medical Information System Program in Medicine, Surgery, and Obstetrics) database

**Gurvan Bourdon**[1]*, **Xavier Lenne**[2], **François Godart**[3], **Laurent Storme**[4], **Didier Theis**[2], **Damien Subtil**[5], **Amelie Bruandet**[2], **Thameur Rakza**[1]

1 Maternity Unit, Department of Gynecology and Obstetrics, Lille University Hospital, Lille, France,
2 Department of Medical Information, Lille University Hospital, Lille, France, 3 Cardiology Unit, Department of Pediatrics, Lille University Hospital, Lille, France, 4 Department of Neonatology, Lille University Hospital, Lille, France, 5 Department of Gynecology and Obstetrics, Lille University Hospital, Lille, France

* gurvan.bourdon@chu-lille.fr

**Data Availability Statement:** Data cannot be shared publicly because access to the PMSI

## Abstract

### Background

Congenital heart defects are common and occur in approximately 0.9% of births. In France, the registries cover approximately 20% of the population but not the entirety of France; therefore, we aimed to update the incidence data for congenital heart defects in France from 2013 to 2022 using the medico-administrative database PMSI-MCO (French Medical Information System Program in Medicine, Surgery, and Obstetrics). We aimed to compare the frequency of risk factors in a population with congenital heart defects and a reference population.

### Methods

From 2013 to 2022, we included children aged < 3 years diagnosed with congenital heart defects according to the International Classification of Diseases, 10th Revision, in the PMSI-MCO database. We compared them with a population without congenital defects on several medical data items (e.g., parity, gemellarity, and mortality rate). Bivariate and multivariate analyses compared children with congenital heart defects and children without congenital malformation.

### Results

We identified 83,879 children with congenital heart defects in France from 2013 to 2022 in the PMSI-MCO database and 7,739,840 children without such defects, including 7,218,952 without any congenital defects. We observed more deaths (7.49% vs. 0.68%, d = 0.59) and more twinning (8.67% vs. 1.23%, d = 0.35) among children with congenital heart defects. Multivariate analysis revealed an increased risk of congenital heart defects in male

database (French Medical Information System Program) is restricted to authorized professionals (with individual access). Likewise, data collected are confidential, only worked data can be disseminated like in our work. To access raw data, one should contact some french professional authorized to access the PMSI (or the institution must request access to the database: demande_base@atih.sante.fr).

**Funding:** The authors received no specific funding for this work.

**Competing interests:** The authors have declared that no competing interests exist.

individuals (OR [odds ratio] 1.056, 95% CI [confidence interval] [1.039–1.076]) and cases of medically assisted reproduction (OR 1.115, 95% CI [1.045–1.189]) and a reduced risk in the case of multiparity (OR 0.921, 95% CI [0.905–0.938]).

## Conclusions

According to the PMSI-MCO database, the incidence of congenital heart defects in France from 2013 to 2022 is 1% of births. Congenital heart defects are more frequent in cases of prematurity, twinning, primiparity, male sex, and maternal age > 40 years.

## Introduction

Congenital defects are common, with an estimated incidence of 25 per 1,000 births, including 30% heart defects according to the European Surveillance of Congenital Anomalies (EURO-CAT) database [1–3]. The incidence of congenital heart defects (CHDs) is increasing, partly because they are better diagnosed, especially since the advent of ultrasonography, and better managed, reducing perinatal deaths and pregnancy termination [4–8]. The overall incidence varies slightly from one region of the world to another but varies according to the CHD considered [4, 9–12]. The life expectancy of patients with CHD is increasing, and its prevalence in adulthood is 4 per 1,000 [13, 14]. This trend calls for regular epidemiological monitoring to adapt healthcare policies.

In France, published epidemiological data are from registers [15–17]. French registries are included in the EUROCAT database [3]. However, these registries are imperfect. An analysis of the systems for monitoring congenital defects was published by the French National Authority for Health (Haute Autorité de Santé [HAS]), which reviewed the strengths and weaknesses of registers [18]. Registries have a good quality of diagnosis but lack completeness in terms of territorial coverage (approximately 20% of France) and do not pool their data. They do not have the same inclusion criteria; for example, one may use the International Statistical Classification of Diseases and Related Health Problems– 10th Revision (ICD-10), whereas another uses a limited list of CHDs [17, 19–21]. In addition, no data are available for some years. Proposals have been made to improve these but this will require time, especially in terms of structuring, harmonizing, and pooling.

There are other databases in France, such as the French Public and Private Hospital Databases with the French Medical Information System Program (*Programme de Médicalisation des Systèmes d'Information* [PMSI]). It is a large national data-base that has shown reliability in numerous epidemiological studies [22–26]. It is used when patients are admitted or hospitalized by coding according to the World Health Organization (WHO) rules and, since 1996, the ICD-10 [20, 21, 27, 28]. The PMSI database collects patients' main and associated diagnoses according to the ICD-10 and sociodemographic data, such as age, sex, and weight. The ICD-10 includes CHD diagnoses and their known risk factors, such as maternal diabetes and trisomy 21 [29, 30]. All information is anonymized using a unique identifier; this avoids duplicates and identifies multiple stays of the same patient, without the possibility of lifting anonymity. For newborns, the PMSI coding is systematic if they are born in a hospital or hospitalized after home delivery. Since 2012, mother-child data have been linked to the PMSI [31].

Our aim was to update French epidemiological data on CHD from 2013 to 2022 using the PMSI database. We also aimed to compare the frequency of the risk factors for CHD in children with CHD and a reference population.

## Materials and methods

Our study followed the French reference methodology MR-005, which regulates access to the PMSI database in accordance with current French Data Protection Authority regulations (Commission Nationale de l'Informatique et des Libertés [CNIL]) [32]. The access to the PMSI database was made available by the French National Agency for the Management of hospitalization data [28]. This was a retrospective, longitudinal, non-interventional study based on an anonymous database. The study did not require patient consent and was registered under CNIL's registration number 2205141.

### Study population and data sources

The inclusion criteria were child < 3 years of age, included in the database between 2013 and 2022 in France, and a diagnosis of CHD according to the ICD-10 (codes Q20 to Q26) [21]. Fetal deaths in utero and pregnancy terminations were also included in the study. The exclusion criteria were as follows: patients residing outside France or with an unspecified place of residence, incorrect French diagnosis-related groups, and incorrect patient identifiers. The reference population consisted of children aged < 3 years between 2013 and 2022 with no diagnosis of congenital defects (absence of any ICD-10 Q code).

PMSI-MCO (médecine-chirurgie-obstétrique [Medicine, Surgery, and Obstetrics]) data were collected between 2013 and 2022. Data from the French Overseas Departments or Regions and Overseas Collectivities (DROM-COM [Départements ou Régions français d'Outre-Mer et les Collectivités d'Outre-Mer]) were analyzed separately. Linking was performed with PMSI-HAD (*hospitalisation à domicile* [home hospitalization]) and mother-child data to increase data reliability.

The CHD diagnoses for ICD-10 codes Q20.0 to Q26.9 were collected, both separately as a single CHD diagnosis and in association with multiple CHD diagnoses. We collected the frequencies of different CHDs.

Medical, social, and demographic data were collected from the patients with CHDs and the reference population. The variables of interest were age at the time of diagnosis, sex, gestational age, birth weight, term of birth, mother's age at birth, parity, route of birth and extraction, whether the child underwent medically assisted reproduction (MAR), death, and age at death.

The data from the ICD-10 codes investigated were: palliative care decision (Z51.5), chromosomal abnormalities (Q90 to Q99), fetus and newborn affected by maternal factors and complications of pregnancy, labor, and delivery (P00 to P04), disorders related to length of gestation and fetal growth (P05 to P08), maternal gestational diabetes (P70.0), maternal diabetes (P70.1), and neonatal withdrawal symptoms from maternal use of drugs of addiction (P96.1). We retained the diagnosis of transposition of the great arteries (TGA) or common arterial trunk in cases of an associated diagnosis of tetralogy of Fallot (ToF) to eliminate some border pathologies.

### Statistical analysis

Categorical variables were presented as absolute numbers and percentages. Continuous variables were presented in classes: $\leq 1$ month and 1–36 months for the age of diagnosis; $\leq 30$ days, 31–365 days, and > 365 days for the days of life; < 24 weeks, 24–36 weeks, and > 36 weeks for the term of birth; < 500 g, 500–1999 g, 2000–3999 g, and $\geq 4000$ g for the birth weight; and $\leq 18$ years, 18–30 years, 31–39 years, and $\geq 40$ years for the maternal age. The chi-square test was used to compare categorical variables. Because the magnitude of statistical significance is heavily influenced by sample size, comparisons between the two groups were

also expressed in terms of the standardized difference score (d, as an absolute number) to provide a more robust and reliable estimation of group divergence [33]. Cohen suggested that d = 0.2, 0.5, and 0.8 represents a small, medium, and large effect size, respectively [34]. Risk factors for CHDs were determined based on their clinical relevance and existing data in the literature using logistic regression. Multiple births were included; therefore, the mothers were included more than once in these situations. A random effects term was introduced to account for these situations. The analyses were performed using the secure platform of the Agence Technique de l'Information Hospitalière. Data extraction and statistical analyses were performed using SAS Guide Enterprise version 8.2. Standardized difference scores were calculated using a specific SAS macro [35].

## Results

### Population

From 2013 to 2022, 88,759 children met the inclusion criteria (83,879 in mainland France and 4,880 in French overseas departments and territories) after excluding erroneous stays and dubious anonymous numbers.

The population without CHD included 8,078,838 children (7,739,840 from metropolitan France and 338,998 from DROM-COM). The reference comparison population without congenital malformations according to ICD-10 codes comprised 7,535,862 children (7,218,952 from mainland France and 316,910 from DROM-COM).

### Epidemiology of CHDs

The incidence of CHDs was 10.7 per 1,000 births in mainland France and 14.2 per 1,000 births in French overseas departments and territories.

The incidence of CHDs for each ICD-10 code in mainland France is shown in Table 1 (incidence of CHDs in DROM-COM in the S1 Table). In cases of association, each code was counted individually; therefore, certain categories of CHDs exceeded 100%. For example, cardiac septal defects accounted for 111.38% of multiple diagnoses, as the codes for ventricular septal defects (VSDs) and atrial septal defects (ASDs) were counted separately, although some children could have had both.

Table 2 shows the frequencies of certain associations in mainland France. The most frequent associations were between VSD and ASD (6.05% of CHD diagnoses), Coarctation of aorta (CoA) and VSD (1.73%), and TGA and VSD (1.46%).

### Comparison of CHD

Table 3 compares children in metropolitan France with CHDs and those without congenital defects (data from the DROM-COM in the S2 Table). A strong relationship of CHD with prematurity (d = 0.86), maternal arterial hypertension (d = 0.26), and twinning (d = 0.35) was observed. Death occurred more frequently among patients with CHD (d = 0.59). Sex and maternal age did not differ between children with CHD and children without congenital malformations in the univariate analysis (d = 0.01 and d = 0.15, respectively).

N: number; P00.0: Fetus and newborn affected by maternal hypertensive disorders; P01.5: Fetus and newborn affected by multiple pregnancy; P05.0: Light for gestational age; P70.0: Syndrome of infant of mother with gestational diabetes; P70.1: Syndrome of infant of a mother with diabetes; P96.1: Neonatal withdrawal symptoms from maternal use of drugs of addiction; d: standardized differences score.

**Table 1. Incidence of congenital heart defects in metropolitan France from 2013 to 2022 in the PMSI-MCO (French Medical Information System Program in Medicine, Surgery and Obstetrics) database according to the ICD-10 (International Statistical Classification of Diseases– 10th Revision).**

| ICD-10 code | | Unique diagnosis | | Multiple diagnoses | | Total | |
|---|---|---|---|---|---|---|---|
| | **Number of patients** | **61,171** | **72.93%** | **22,708** | **27.07%** | **83,879** | **100.00%** |
| **Q20** | **Congenital malformations of cardiac chambers and connections** | **1,894** | **3.10%** | **6,261** | **27.57%** | **8,155** | **9.72%** |
| Q200 | Common arterial trunk | 127 | 0.21% | 313 | 1.38% | 440 | 0.52% |
| Q201 | Double outlet right ventricle | 62 | 0.10% | 960 | 4.23% | 1,022 | 1.22% |
| Q202 | Double outlet left ventricle | 12 | 0.02% | 102 | 0.45% | 114 | 0.14% |
| Q203 | Discordant ventriculoarterial connection | 666 | 1.09% | 2,143 | 9.44% | 2,809 | 3.35% |
| Q204 | Double inlet ventricle | 92 | 0.15% | 977 | 4.30% | 1,069 | 1.27% |
| Q205 | Discordant atrioventricular connection | 36 | 0.06% | 449 | 1.98% | 485 | 0.58% |
| Q206 | Isomerism of atrial appendages | 18 | 0.03% | 126 | 0.55% | 144 | 0.17% |
| Q208 | Other congenital malformations of cardiac chambers and connections | 652 | 1.07% | 759 | 3.34% | 1,411 | 1.68% |
| Q209 | Congenital malformation of cardiac chambers and connections, unspecified | 229 | 0.37% | 432 | 1.90% | 661 | 0.79% |
| **Q21** | **Congenital malformations of cardiac septa** | **28,724** | **46.96%** | **25,293** | **111.38%** | **54,017** | **64.40%** |
| Q210 | Ventricular septal defect | 13,753 | 22.48% | 10,811 | 47.61% | 24,564 | 29.29% |
| Q211 | Atrial septal defect | 12,187 | 19.92% | 10,138 | 44.65% | 22,325 | 26.62% |
| Q212 | Atrioventricular septal defect | 887 | 1.45% | 1,961 | 8.64% | 2,848 | 3.40% |
| Q213 | Tetralogy of Fallot | 1,398 | 2.29% | 1,838 | 8.09% | 3,236 | 3.86% |
| Q214 | Aortopulmonary septal defect | 58 | 0.09% | 183 | 0.81% | 241 | 0.29% |
| Q218 | Other congenital malformations of cardiac septa | 377 | 0.62% | 291 | 1.28% | 668 | 0.80% |
| Q219 | Congenital malformation of cardiac septum, unspecified | 64 | 0.10% | 71 | 0.31% | 135 | 0.16% |
| **Q22** | **Congenital malformations of pulmonary and tricuspid valves** | **1,758** | **2.87%** | **5,520** | **24.31%** | **7,278** | **8.68%** |
| Q220 | Pulmonary valve atresia | 114 | 0.19% | 959 | 4.22% | 1,073 | 1.28% |
| Q221 | Congenital pulmonary valve stenosis | 1,088 | 1.78% | 1,976 | 8.70% | 3,064 | 3.65% |
| Q222 | Congenital pulmonary valve insufficiency | 30 | 0.05% | 262 | 1.15% | 292 | 0.35% |
| Q223 | Other congenital malformations of pulmonary valve | 108 | 0.18% | 519 | 2.29% | 627 | 0.75% |
| Q224 | Congenital tricuspid stenosis | 42 | 0.07% | 436 | 1.92% | 478 | 0.57% |
| Q225 | Ebstein anomaly | 144 | 0.24% | 185 | 0.81% | 329 | 0.39% |
| Q226 | Hypoplastic right heart syndrome | 36 | 0.06% | 495 | 2.18% | 531 | 0.63% |
| Q228 | Other congenital malformations of tricuspid valve | 167 | 0.27% | 567 | 2.50% | 734 | 0.88% |
| Q229 | Congenital malformation of tricuspid valve, unspecified | 29 | 0.05% | 121 | 0.53% | 150 | 0.18% |
| **Q23** | **Congenital malformations of aortic and mitral valves** | **1,352** | **2.21%** | **5,186** | **22.84%** | **6,538** | **7.79%** |
| Q230 | Congenital stenosis of aortic valve | 179 | 0.29% | 788 | 3.47% | 967 | 1.15% |
| Q231 | Congenital insufficiency of aortic valve | 362 | 0.59% | 1,216 | 5.35% | 1,578 | 1.88% |
| Q232 | Congenital mitral stenosis | 21 | 0.03% | 624 | 2.75% | 645 | 0.77% |
| Q233 | Congenital mitral insufficiency | 209 | 0.34% | 833 | 3.67% | 1,042 | 1.24% |
| Q234 | Hypoplastic left heart syndrome | 409 | 0.67% | 1,059 | 4.66% | 1,468 | 1.75% |
| Q238 | Other congenital malformations of aortic and mitral valves | 131 | 0.21% | 523 | 2.30% | 654 | 0.78% |
| Q239 | Congenital malformation of aortic and mitral valves, unspecified | 41 | 0.07% | 143 | 0.63% | 184 | 0.22% |
| **Q24** | **Other congenital malformations of heart** | **4,633** | **7.57%** | **7,334** | **32.30%** | **11,967** | **14.27%** |
| Q240 | Dextrocardia | 134 | 0.22% | 305 | 1.34% | 439 | 0.52% |
| Q241 | Levocardia | 178 | 0.29% | 480 | 2.11% | 658 | 0.78% |
| Q242 | Cor triatriatum | 27 | 0.04% | 69 | 0.30% | 96 | 0.11% |
| Q243 | Pulmonary infundibular stenosis | 21 | 0.03% | 501 | 2.21% | 522 | 0.62% |
| Q244 | Congenital subaortic stenosis | 49 | 0.08% | 429 | 1.89% | 478 | 0.57% |
| Q245 | Malformation of coronary vessels | 308 | 0.50% | 549 | 2.42% | 857 | 1.02% |
| Q246 | Congenital heart block | 128 | 0.21% | 56 | 0.25% | 184 | 0.22% |
| Q248 | Other specified congenital malformations of heart | 2,351 | 3.84% | 2,422 | 10.67% | 4,773 | 5.69% |

*(Continued)*

**Table 1.** (Continued)

| ICD-10 code | | Unique diagnosis | | Multiple diagnoses | | Total | |
|---|---|---|---|---|---|---|---|
| Q249 | Congenital malformation of heart, unspecified | 1,437 | 2.35% | 2,523 | 11.11% | 3,960 | 4.72% |
| **Q25** | **Congenital malformations of great arteries** | **22,028** | **36.01%** | **16,488** | **72.61%** | **38,516** | **45.92%** |
| Q250 | Patent ductus arteriosus | 18,398 | 30.08% | 6,249 | 27.52% | 24,647 | 29.38% |
| Q251 | Coarctation of aorta | 1,363 | 2.23% | 2,913 | 12.83% | 4,276 | 5.10% |
| Q252 | Atresia of aorta | 24 | 0.04% | 399 | 1.76% | 423 | 0.50% |
| Q253 | Stenosis of aorta | 104 | 0.17% | 482 | 2.12% | 586 | 0.70% |
| Q254 | Other congenital malformations of aorta | 844 | 1.38% | 1,872 | 8.24% | 2,716 | 3.24% |
| Q255 | Atresia of pulmonary artery | 59 | 0.10% | 802 | 3.53% | 861 | 1.03% |
| Q256 | Stenosis of pulmonary artery | 678 | 1.11% | 1,938 | 8.53% | 2,616 | 3.12% |
| Q257 | Other congenital malformations of pulmonary artery | 198 | 0.32% | 726 | 3.20% | 924 | 1.10% |
| Q258 | Other congenital malformations of great arteries | 262 | 0.43% | 689 | 3.03% | 951 | 1.13% |
| Q259 | Congenital malformation of great arteries, unspecified | 98 | 0.16% | 418 | 1.84% | 516 | 0.62% |
| **Q26** | **Congenital malformations of great veins** | **782** | **1.28%** | **2,373** | **10.45%** | **3,155** | **3.76%** |
| Q260 | Congenital stenosis of vena cava | 4 | 0.01% | 19 | 0.08% | 23 | 0.03% |
| Q261 | Persistent left superior vena cava | 172 | 0.28% | 464 | 2.04% | 636 | 0.76% |
| Q262 | Total anomalous pulmonary venous connection | 114 | 0.19% | 437 | 1.92% | 551 | 0.66% |
| Q263 | Partial anomalous pulmonary venous connection | 40 | 0.07% | 370 | 1.63% | 410 | 0.49% |
| Q264 | Anomalous pulmonary venous connection, unspecified | 42 | 0.07% | 439 | 1.93% | 481 | 0.57% |
| Q265 | Anomalous portal venous connection | 13 | 0.02% | 16 | 0.07% | 29 | 0.03% |
| Q266 | Portal vein-hepatic artery fistula | 94 | 0.15% | 63 | 0.28% | 157 | 0.19% |
| Q268 | Other congenital malformations of great veins | 267 | 0.44% | 501 | 2.21% | 768 | 0.92% |
| Q269 | Congenital malformation of great vein unspecified | 36 | 0.06% | 64 | 0.28% | 100 | 0.12% |

In the multivariate analysis, maternal age > 40 years was a risk factor for CHD (OR 1.310, p < 0.001) (Fig 1 for metropolitan France; data in S1 Fig for DROM-COM). Similarly, male sex and pregnancy resulting from MAR were associated with an increased risk of CHD (OR, 1.057 and 1.115, respectively; p < 0.001), whereas multiparity had a protective effect against CHD (OR 0.921, p < 0.001).

## Discussion

### Interpretation of results

To our knowledge, this is the first study to calculate the incidence of CHDs in France using the PMSI database. Its aim was not to replace the registries but to study the CHD diagnoses in France using another reference system to increase its completeness.

Studies suggest that the PMSI database is an appropriate data source for epidemiological studies [22–26]. Our methodology using the PMSI database seems correct because we found the same incidences as those reported in the literature [2, 4, 16, 17]. Therefore, a comparison between populations with CHD and those without congenital malformations is legitimate. We excluded children with other congenital malformations from the reference population to avoid confounding factors between the data studied. It is important to note that the number of children varied according to the secondary criteria studied, because information was not always available.

The PMSI database provided an exhaustive list of CHD diagnoses in France, as coding is compulsory and is carried out at the time of diagnosis. This would not have been possible with registries [18]. The population studied was large, increasing the relevance of the comparison

**Table 2. Associations between congenital heart defect diagnoses in metropolitan France from 2013 to 2022 in the PMSI-MCO (French Medical Information System Program in Medicine, Surgery and Obstetrics) database.**

| First ICD-10 code | | Second ICD-10 code | | Number of patients | Percentage of patients |
|---|---|---|---|---|---|
| Q203 | Discordant ventriculoarterial connection | Q201 | Double outlet right ventricle | 349 | 0.42% |
| Q203 | Discordant ventriculoarterial connection | Q202 | Double outlet left ventricle | 42 | 0.05% |
| Q203 | Discordant ventriculoarterial connection | Q210 | Ventricular septal defect | 1,221 | 1.46% |
| Q203 | Discordant ventriculoarterial connection | Q251 | Coarctation of aorta | 276 | 0.33% |
| Q203 | Discordant ventriculoarterial connection | Q255 | Atresia of pulmonary artery | 86 | 0.10% |
| Q203 | Discordant ventriculoarterial connection | Q256 | Stenosis of pulmonary artery | 281 | 0.34% |
| Q210 | Ventricular septal defect | Q211 | Atrial septal defect | 5,073 | 6.05% |
| Q210 | Ventricular septal defect | Q220 | Pulmonary valve atresia | 442 | 0.53% |
| Q210 | Ventricular septal defect | Q251 | Coarctation of aorta | 1,455 | 1.73% |
| Q210 | Ventricular septal defect | Q255 | Atresia of pulmonary artery | 443 | 0.53% |
| Q213 | Tetralogy of Fallot | Q251 | Coarctation of aorta | 16 | 0.02% |
| Q220 | Pulmonary valve atresia | Q224 | Congenital tricuspid stenosis | 107 | 0.13% |
| Q220 | Pulmonary valve atresia | Q225 | Ebstein anomaly | 21 | 0.03% |
| Q221 | Congenital pulmonary valve stenosis | Q224 | Congenital tricuspid stenosis | 59 | 0.07% |
| Q221 | Congenital pulmonary valve stenosis | Q225 | Ebstein anomaly | 12 | 0.01% |
| Q220 | Pulmonary valve atresia | Q224 | Congenital tricuspid stenosis | 143 | 0.17% |
| Q220 | Pulmonary valve atresia | Q225 | Ebstein anomaly | 71 | 0.08% |
| Q221 | Congenital pulmonary valve stenosis | Q224 | Congenital tricuspid stenosis | 1,455 | 1.73% |
| Q221 | Congenital pulmonary valve stenosis | Q225 | Ebstein anomaly | 212 | 0.25% |
| Q230 | Congenital stenosis of aortic valve | Q232 | Congenital mitral stenosis | 89 | 0.11% |
| Q230 | Congenital stenosis of aortic valve | Q233 | Congenital mitral insufficiency | 79 | 0.09% |
| Q251 | Coarctation of aorta | Q210 | Ventricular septal defect | 12 | 0.01% |
| Q251 | Coarctation of aorta | Q232 | Congenital mitral stenosis | 56 | 0.07% |
| Q251 | Coarctation of aorta | Q233 | Congenital mitral insufficiency | 26 | 0.03% |
| Q252 | Atresia of aorta | Q232 | Congenital mitral stenosis | 81 | 0.10% |
| Q252 | Atresia of aorta | Q233 | Congenital mitral insufficiency | 185 | 0.22% |
| Q253 | Stenosis of aorta | Q232 | Congenital mitral stenosis | 44 | 0.05% |
| Q253 | Stenosis of aorta | Q233 | Congenital mitral insufficiency | 349 | 0.42% |
| Q262 | Total anomalous pulmonary venous connection | Q210 | Ventricular septal defect | 42 | 0.05% |
| Q262 | Total anomalous pulmonary venous connection | Q211 | Atrial septal defect | 1,221 | 1.46% |
| Q262 | Total anomalous pulmonary venous connection | Q212 | Atrioventricular septal defect | 276 | 0.33% |

between the CHD and unaffected populations on numerous criteria, which the PMSI database also makes possible.

However, unlike registries, what we gained in exhaustiveness may be lost in diagnostic precision. ICD-10 codes are not very precise (e.g., Q24.9 is "congenital malformation of the heart, unspecified"). The PMSI database is also at risk of coding errors when not carried out by specialists or by poorly trained healthcare professionals, albeit improving [25, 28, 36]. ICD-10 coding is compulsory; however, some ICD-10 codes have no impact on the cost of hospital stays and thus may not be coded. We included children under 3 years of age to limit incomplete data. The PMSI database does not allow searching for more than 10 years back in time, therefore if we had included older children or even adults, recovering the data would not have been possible. This should not have decreased our exhaustiveness because CHDs are diagnosed early in a child's life and even more so if they are severe [4]. We chose the place of diagnosis and not the place of birth or residence to establish a CHD diagnosis in France. This choice appeared to have the lowest risk of child misallocation. We were unable to distinguish between prenatal and postnatal CHD diagnoses.

**Table 3. Frequency of perinatal data between children with congenital heart defects and children without congenital malformation in metropolitan France from 2013 to 2022.**

| Studied factor | | | Patients with congenital heart defects | | Patients without congenital malformation | | d |
|---|---|---|---|---|---|---|---|
| Number/percentage of patients | | | 83,879 | 100.00% | 7,218,952 | 100.00% | - |
| Age at diagnosis (month) | | [0–1] | 66,687 | 79.50% | - | - | - |
| | | [1–36] | 17,192 | 20.50% | - | - | - |
| Death (days of life) | | N | 6,282 | 7.49% | 49,308 | 0.68% | 0.59 |
| | | ≤ 30 | 4,745 | 75.53% | 46,652 | 94.61% | |
| | | 31–365 | 1,302 | 20.73% | 1,584 | 3.21% | |
| | | > 365 | 235 | 3.74% | 1,072 | 2.17% | |
| Palliative care | | Yes | 1,691 | 2.02% | 2,529 | 0.04% | 0.20 |
| | | No | 82,188 | 97.98% | 7,216,423 | 99.96% | |
| Sex | | Male | 43,523 | 51.89% | 3,718,349 | 51.51% | 0.01 |
| | | Female | 40,356 | 48.11% | 3,500,603 | 48.49% | |
| Term of birth (gestation week) | | < 24 | 448 | 0.74% | 11,586 | 0.19% | 0.86 |
| | | 24–36 | 21,608 | 35.89% | 304,826 | 5.13% | |
| | | > 36 | 38,154 | 63.37% | 5,630,301 | 94.68% | |
| Birth weight (g) | | < 500 | 284 | 0.47% | 8,718 | 0.15% | 0.78 |
| | | 500–1999 | 16,793 | 27.89% | 96,976 | 1.63% | |
| | | 2000–3999 | 40,019 | 66.47% | 5,419,024 | 91.13% | |
| | | ≥ 4000 | 3,114 | 5.17% | 421,995 | 7.10% | |
| Mother's age at birth (years) | | < 18 | 275 | 0.52% | 23,560 | 0.41% | 0.15 |
| | | 18–30 | 26,778 | 50.41% | 2,936,870 | 51.28% | |
| | | 31–39 | 22,749 | 42.82% | 2,517,843 | 43.96% | |
| | | ≥ 40 | 3,319 | 6.25% | 248,820 | 4.34% | |
| Mother's parity | | Primiparity | 29,874 | 56.24% | 2,830,423 | 49.42% | 0.14 |
| | | Multiparity | 23,247 | 43.76% | 2,896,670 | 50.58% | |
| Birth mode | Mode of delivery | Cesarean | 18,534 | 34.89% | 1,089,582 | 19.03% | 0.36 |
| | | Vaginal birth | 34,587 | 65.11% | 4,637,511 | 80.97% | |
| | Instrumental birth | Yes | 4,849 | 9.13% | 645,708 | 11.27% | 0.07 |
| | | No | 48,272 | 90.87% | 5,081,385 | 88.73% | |
| Medically assisted reproduction | | Yes | 1,013 | 1.91% | 80,990 | 1.41% | 0.04 |
| | | No | 52,108 | 98.09% | 5,646,103 | 98.59% | |
| Perinatal anomalies | | P00.0 | 5,643 | 6.73% | 111,196 | 1.54% | 0.26 |
| | | P01.5 | 7,273 | 8.67% | 88,592 | 1.23% | 0.35 |
| | | P05.0 | 10,713 | 12.77% | 269,632 | 3.74% | 0.33 |
| | | P70.0 | 5,501 | 6.56% | 326,018 | 4.52% | 0.09 |
| | | P70.1 | 1,592 | 1.90% | 39,009 | 0.54% | 0.12 |
| | | P96.1 | 284 | 0.34% | 6,493 | 0.09% | 0.05 |

DROM-COM is separate from the metropolitan population because the populations are different and access to diagnostic and care facilities is sometimes more difficult [37]. Overall, we found the same differences between the populations and similar risk factors as those observed in the mainland population.

## Comparisons with the literature

The incidences found in this study were fairly identical to those reported in the literature for each type of CHD, particularly TGA, ToF, and CoA [2, 4]. However, we have observed more

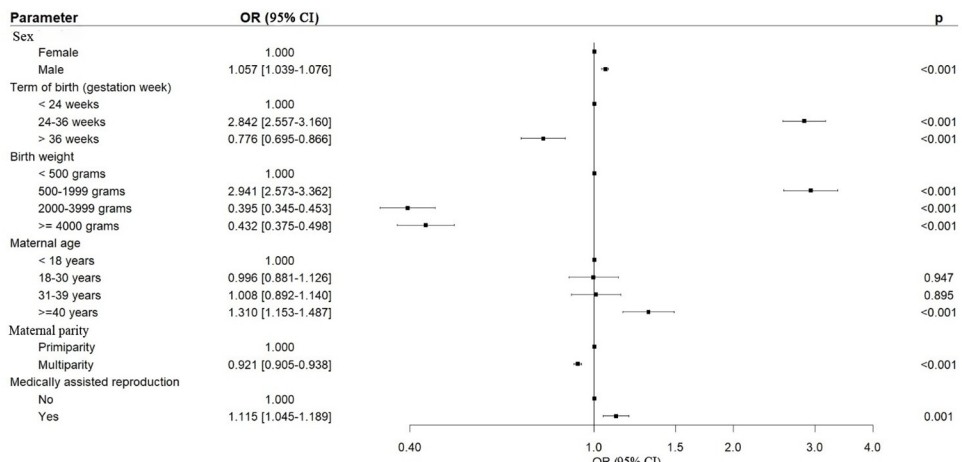

**Fig 1. Results of multivariate analysis in metropolitan France.** Children with congenital heart defects are compared with the reference population from 2013 to 2022. OR: odds ratio; CI: confidence interval.

cases of patent ductus arteriosus. This difference could be due to the title of the coding; ductus arteriosus is coded as soon as it is present, particularly in premature babies, where it is extremely common without always impacting the child [5, 38]. However, if all diagnoses of patent ductus arteriosus were excluded (i.e., exclusion in excess), the incidence of CHD would be 8.5%, remaining fairly identical to other studies [3–5].

This partly explains why premature birth and low birth weight, which may be associated with CHD, emerged as risk factors for CHD. However, prematurity appears to be a risk factor for CHD independent of ductus arteriosus. Chu et al. showed an overall increase in the incidence of CHD in children born prematurely between 25 and 32 weeks of gestation, with a high percentage of severe malformations, excluding children with a persistent ductus arteriosus [39].

Much of our data are consistent with what has been described previously. CHDs are a major cause of early death in children [1]. Most diagnoses are made before 1 month of life, including antenatal diagnoses [17]. There were more caesarean sections in cases of CHD, with no differences depending on the type of CHD [40, 41]. MAR is associated with a higher risk of CHD [42]. In twins, there was an increased risk of CHD [43].

Some discrepancies between our data and those published in literatures contribute to the discussion of the inconsistency of the literature. In our study, maternal age > 40 years was a risk factor in multivariate analysis, whereas maternal age > 35 years was sometimes described as a risk factor but not always [11, 29, 44, 45]. Similarly, male individuals were slightly more at risk of CHD in our study, which is consistent with the results of some studies but not others [11, 45–47].

Some of the associations described were either absent or weak. We observed fewer chromosomal abnormalities in children with CHD; however, they remained more numerous than those in the general population [48, 49]. One explanation could be the choice of using ICD-10 codes Q90 to Q99, which does not include all chromosomal and genetic abnormalities but is the easiest to research. This deviation could also be due to the lack of precision in the ICD-10 codes. Similarly, maternal diabetes did not appear to be a risk factor for CHD in our study; however, we only studied diabetes declared during pregnancy, unlike other authors [50].

According to our results, there were no excess mortality in cases of TGA, ToF, or CoA when these were considered serious CHDs, contrary to what has been previously described

[51–53]. This may be explained by improved diagnosis and management of these diseases, which have become standardized.

Contrary to our study, in which multiparity appeared to be protective, Feng et al. showed in their meta-analysis an increased risk of CHD in repeated pregnancies. However, their results were affected by the inclusion of heterogeneous studies and showed an absence of statistical difference when directly comparing primiparous and multiparous women [54].

### Improved data collection

The accuracy of the PMSI database could be improved using the ICD– 11th Revision published in 2018 by the WHO, which codes CHDs using a simplified version of the International Pediatric and Congenital Cardiac Code [55, 56]. However, this precise CHD classification has not yet been applied in France.

It seems worthwhile to use the data from the PMSI to feed the registers. Despite these limitations, registers remain a valid solution that avoids most coding accuracy problems [18]. The database already exists and is classified into most registries. In a French study led by the HAS on registers, it was suggested that existing registers should work together more closely to populate a common database rather than setting up a single register (which would incur more cost and human resources and would lack proximity).

## Conclusions

The incidence of CHDs in metropolitan France from 2013 to 2022 is 10.7 per 1,000 births, according to the PMSI database. The PMSI database can be used in conjunction with registries to establish the most complete epidemiology of CHDs in France. CHDs are more frequent in cases of prematurity, twinning, primiparity, male sex, and maternal age of > 40 years.

## Supporting information

**S1 Table. Incidence of congenital heart defect in non-metropolitan France from 2013 to 2022 in the PMSI-MCO (French Medical Information System Program in Medicine, Surgery and Obstetrics) database according to the ICD-10 (International Statistical Classification of Diseases– 10th Revision).**
(DOCX)

**S2 Table. Frequency of perinatal data between children with congenital heart defects and children without congenital malformation in non-metropolitan France from 2013 to 2022.**
(DOCX)

**S1 Fig. Results of multivariate analysis in non metropolitan France.** Children with congenital heart defects are compared with the reference population from 2013 to 2022. OR: odds ratio; CI: confidence interval.
(TIF)

## Author Contributions

**Conceptualization:** Gurvan Bourdon, Laurent Storme, Amelie Bruandet, Thameur Rakza.

**Data curation:** Xavier Lenne, Amelie Bruandet.

**Formal analysis:** Xavier Lenne.

**Methodology:** Gurvan Bourdon, Xavier Lenne, Damien Subtil, Amelie Bruandet.

**Project administration:** Didier Theis.

**Supervision:** Didier Theis, Thameur Rakza.

**Validation:** Gurvan Bourdon, Amelie Bruandet.

**Visualization:** Gurvan Bourdon, François Godart, Damien Subtil, Amelie Bruandet, Thameur Rakza.

**Writing – original draft:** Gurvan Bourdon.

**Writing – review & editing:** Gurvan Bourdon, Xavier Lenne, François Godart, Laurent Storme, Didier Theis, Damien Subtil, Amelie Bruandet, Thameur Rakza.

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
