## [Decision Letter · Decision Letter 0]

21 Sep 2023

PONE-D-23-20982French epidemiology of congenital heart defects from 2013 to 2022 using the PMSI-MCO (French Medical Information System Program in Medicine, Surgery and Obstetrics) DatabasePLOS ONE

Dear Dr. BOURDON,

Thank you for submitting your manuscript to PLOS ONE. After careful consideration, we feel that it has merit but does not fully meet PLOS ONE’s publication criteria as it currently stands. Therefore, we invite you to submit a revised version of the manuscript that addresses the points raised during the review process.

ACADEMIC EDITOR: In addition to addressing the comments given by the reviewers, please address the following issues as well.The study aim that is mentioned in the abstract section is different from the one mentioned at the end of the introduction section.  Methods: Exclusion/inclusion criteria is not clear. Why children greater than 2yrs old were excluded? Can population disease incidence be accurate without including all age groups who have the disease? The study is French Epidemiology of CHD! Also, data extraction method employed needs further elaboration.Result: Findings in the result and discussion sections are based on data only from Metropolitan France while the method section states that data were collected from French Overseas territories as well.Tables in the result section are lengthy and difficult to understand.Conclusion is not based the findings in the result section. Eg. Authors concluded by saying “The PMSI data base is reliable…” while they didn’t assess reliability of PMSI data base.==============================

We look forward to receiving your revised manuscript.

Kind regards,

Atnafu Mekonnen Tekleab, M.D

Academic Editor

PLOS ONE

Reviewers' comments:

Reviewer's Responses to Questions

**Comments to the Author**

1. Is the manuscript technically sound, and do the data support the conclusions?

Reviewer #1: No

Reviewer #2: Partly

2. Has the statistical analysis been performed appropriately and rigorously? 

Reviewer #1: No

Reviewer #2: I Don't Know

3. Have the authors made all data underlying the findings in their manuscript fully available?

Reviewer #1: No

Reviewer #2: Yes

4. Is the manuscript presented in an intelligible fashion and written in standard English?

Reviewer #1: No

Reviewer #2: Yes

5. Review Comments to the Author

Reviewer #1: Abstract

The authors have written abstracts very well. However, the PMSI-MCO abbreviation does not match with the French Medical Information System Program in Medicine, Surgery, and Obstetrics.

Introduction

The first two paragraphs of the introduction are written appropriately. However, the rest of the introduction paragraphs are written very poorly. Authors need to paraphrase whole paragraphs 3,4 and 5. The authors have described flaws in the database but need to describe the strength of PMSI as well to support why authors have used PMSI.

Method

What is reference methodology MR-005? Please describe in detail.

Please explain why this study has an anonymous database and does not involve human subjects. It seems like authors know database and database does have human subject information. Please clarify this.

Study population and data sources

Were there confirmed diagnoses of CHD among fetal death and termination of pregnancy cases?

Why did the authors need to collect and analyze data from the French Overseas Departments or regions and Overseas collectivities?

On page 5, line 124: It should frequency of different CHDs. There should not be an association of different CHDs.

On page 6, line 135: What does it mean to prioritize the diagnosis of TGA and ToF in the study?

Statistical analysis

The authors presented data on the children’s age at the time of diagnosis. In their study population they had included children who were <= 2 years of age however, in their results section authors presented data of children who were diagnosed with CHD up to 3 years of age.

Per the authors, multiple births were included so mothers were included more than once which makes maternal data analyses completely wrong due to double counting.

Page 7, line 151: for above reference.

Table 2: Represent association of CHD however, it seems like the authors did not show any appropriate statistical analyses to present the association of CHD.

Overall, the Manuscript demonstrates lots of grammatical and improper formation of sentences. The study design and method are not adequately formed to present epidemiological data. There is a huge discrepancy in inclusion criteria and study results which means this study is not designed and presented correctly. This study needs thorough revision in design, study population, and analysis in the future.

Reviewer #2: (1) I would suggest author to use Congenital Heart Defect instead of disease in through out paper to keep it consistent. both are appropriate Synonyms in ICD10 coding

(2) Page 4 Line 77-78, needs revision with more specific message to readers

(3) Page 4 Line 80-82, could you elaborate on sentences you have mentioned instead of general statement

(4) Introduction: I would recommended author for few revision with grammatical correction and more professional language use in few places like "It is used whenever a patient is hospitalized" instead of that author should use "it is used when patient is admitted or hospitalized"

" We wanted to compare the frequency of" instead of that "Our aim of the study is to compare the frequency of"

(5) Does author exclude patient with transfer to other hospital data? or readmission ? could you mention in exclusion criteria?

(6) Page 13, Line 224-225, Any studies was done by using similar database? I would like author to put study who have used similar database to authenticate the method for data extraction and secondary analysis of database.

(7) Line 242-243, can you elaborate on this sentence "We included children under 2

years of age in order to limit side effects and incomplete data."

(8) Author should discuss what is limitation of database and what different studies in past have been done using PMSI database.

6. PLOS authors have the option to publish the peer review history of their article (what does this mean?). If published, this will include your full peer review and any attached files.

Reviewer #1: No

Reviewer #2: No

---

## [Author Response · Author response to Decision Letter 0]

10 Jan 2024

Thank you for the constructive comments regarding my article. This time I used a professional english editing, I hope your reading will be more enjoyable.

Response to Reviewers and academic editor :

Academic editor: 

• The study aim that is mentioned in the abstract section is different from the one mentioned at the end of the introduction section.

->Corrected in the text 

• Methods: Exclusion/inclusion criteria is not clear. Why children greater than 2yrs old were excluded? Can population disease incidence be accurate without including all age groups who have the disease? The study is French Epidemiology of CHD! Also, data extraction method employed needs further elaboration.

-> Addition in the text (in the discussion to the age of children, to minimize the lack of date).

• Result: Findings in the result and discussion sections are based on data only from Metropolitan France while the method section states that data were collected from French Overseas territories as well.

->It’s epidemiology in France so we collected data in metropolitan and overseas territories. But it’s not exactly the same population so we divided the analyses. So as not to overload the already heavy tables we show the overseas territories data at supplementary data.

• Tables in the result section are lengthy and difficult to understand.

->I do not have an obvious solution… I have tried some modification.

• Conclusion is not based the findings in the result section. Eg. Authors concluded by saying “The PMSI data base is reliable…” while they didn’t assess reliability of PMSI data base.

->Corrected in the text 

Reviewer #1:

• Abstract

The authors have written abstracts very well. However, the PMSI-MCO abbreviation does not match with the French Medical Information System Program in Medicine, Surgery, and Obstetrics.

->True, it was the French abbreviation (Surgery = Chirurgie). Add in the text (in addition to the title), but not to the abstract.

• Introduction

The first two paragraphs of the introduction are written appropriately. However, the rest of the introduction paragraphs are written very poorly. Authors need to paraphrase whole paragraphs 3,4 and 5. The authors have described flaws in the database but need to describe the strength of PMSI as well to support why authors have used PMSI.

->Using a professional English editing.

We added references about the reliability of using the PMSI in epidemiologic studies.

• Method

What is reference methodology MR-005? Please describe in detail.

Please explain why this study has an anonymous database and does not involve human subjects. It seems like authors know database and database does have human subject information. Please clarify this.

->Methodology MR-005 is a French classification about research. MR-001 is for example an experimental research on human using a new medicine or in new indication. MR-005 is a methodology specially created for access to PMSI, a national anonymous database where is not possible to lift anonymity. It does not require patient’s authorization, but everybody could oppose to the use of their hospitalization data for studies. Could only have access to the PMSI health establishments and hospital federations.

Effectively, PMSI involve human subjects or more precisely human data… That is an error on our part, corrected in the text.

• Study population and data sources

Were there confirmed diagnoses of CHD among fetal death and termination of pregnancy cases?

->These diagnoses were included in the groups “age at diagnosis < 1 month” and “death < 30 days”. They could obviously be associated with others congenital defects.

• Why did the authors need to collect and analyze data from the French Overseas Departments or regions and Overseas collectivities?

->It’s epidemiology in France so we collected data in metropolitan and overseas territories. But it’s not exactly the same population so we divided the analyses. So as not to overload the already heavy tables we show the overseas territories data at supplementary data.

• On page 5, line 124: It should frequency of different CHDs. There should not be an association of different CHDs.

->Corrected in the text 

• On page 6, line 135: What does it mean to prioritize the diagnosis of TGA and ToF in the study?

->Addition in the text. In some case, it is possible to have a TGA with an overriding aorta and a pulmonary stenosis. The PMSI is not always completed by the cardiologist and some patients could have in this situation both codes in the database despise being only one pathology.

• Statistical analysis

The authors presented data on the children’s age at the time of diagnosis. In their study population they had included children who were <= 2 years of age however, in their results section authors presented data of children who were diagnosed with CHD up to 3 years of age.

->Corrected in the text. It was 2 years included, but to not confused we modified the text.

• Per the authors, multiple births were included so mothers were included more than once which makes maternal data analyses completely wrong due to double counting.

Page 7, line 151: for above reference.

->Taken into account in the statistical analysis

• Table 2: Represent association of CHD however, it seems like the authors did not show any appropriate statistical analyses to present the association of CHD.

->This is the raw incidence of associations. What type of analysis are you talking about?

Overall, the Manuscript demonstrates lots of grammatical and improper formation of sentences. The study design and method are not adequately formed to present epidemiological data. There is a huge discrepancy in inclusion criteria and study results which means this study is not designed and presented correctly. This study needs thorough revision in design, study population, and analysis in the future.

Reviewer #2:

• I would suggest author to use Congenital Heart Defect instead of disease in through out paper to keep it consistent. both are appropriate Synonyms in ICD10 coding

->Corrected in the text.

• Page 4 Line 77-78, needs revision with more specific message to readers

->Addition in the text

• Page 4 Line 80-82, could you elaborate on sentences you have mentioned instead of general statement

->Addition in the text

• Introduction: I would recommended author for few revision with grammatical correction and more professional language use in few places like "It is used whenever a patient is hospitalized" instead of that author should use "it is used when patient is admitted or hospitalized"

" We wanted to compare the frequency of" instead of that "Our aim of the study is to compare the frequency of"

->Using a professional English editing.

• Does author exclude patient with transfer to other hospital data? or readmission ? could you mention in exclusion criteria?

->Add to the description of the PMSI (in introduction). Each patient has a unique social security number, which generates a unique PMSI code. So multiple inclusion of the same patient is not possible if care during the analyses.

• Page 13, Line 224-225, Any studies was done by using similar database? I would like author to put study who have used similar database to authenticate the method for data extraction and secondary analysis of database.

->Add to the discussion.

• Line 242-243, can you elaborate on this sentence "We included children under 2

years of age in order to limit side effects and incomplete data."

->Add to the discussion.

• Author should discuss what is limitation of database and what different studies in past have been done using PMSI database.

->Slightly add to the discussion.

---

## [Editor Report · Decision Letter 1]

22 Jan 2024

Epidemiology of congenital heart defects in France from 2013 to 2022 using the PMSI-MCO (French Medical Information System Program in Medicine, Surgery, and Obstetrics) database

PONE-D-23-20982R1

Dear Dr. Bourdon,

We’re pleased to inform you that your manuscript has been judged scientifically suitable for publication and will be formally accepted for publication once it meets all outstanding technical requirements.

Kind regards,

Atnafu Mekonnen Tekleab, M.D

Academic Editor

PLOS ONE
---

## [Editor Report · Acceptance letter]

15 Feb 2024

PONE-D-23-20982R1 

PLOS ONE

Dear Dr. BOURDON, 

I'm pleased to inform you that your manuscript has been deemed suitable for publication in PLOS ONE. Congratulations! Your manuscript is now being handed over to our production team.

Kind regards, 

on behalf of

Dr. Atnafu Mekonnen Tekleab 

Academic Editor

PLOS ONE